

# The importance of spatial resolution in the modelling of methane emissions from natural wetlands

Yousef Albuhaisi[1], Ype van der Velde[1] and Sander Houweling[1] [2]

[1]Vrije Universiteit, Department of Earth Sciences, Amsterdam, the Netherlands,
[2]SRON Netherlands Institute for Space Research, Utrecht, the Netherlands,

*Correspondence to*: Yousef Albuhaisi (y.albuhaisi@vu.nl)

**Abstract.**

An important uncertainty in the modelling of methane ($CH_4$) emissions from natural wetlands is the wetland area. It is
difficult to model wetlands $CH_4$ emissions because of several factors, including its spatial heterogeneity on a large range
of scales. In this study, we investigate the impact of model resolution on the simulated wetland methane emission for the
Fenno-Scandinavian Peninsula. This is done using a high-resolution wetland map ($100x100m^2$) and soil carbon map
($250x250m^2$) in combination with a highly simplified $CH_4$ emission model that is coarsened in six steps from 0.005° to 1°.
We find a strong relation between wetland emissions and resolution, which is sensitive, however, to the sub-grid treatment
of the wetland fraction. In our setup, soil carbon and soil moisture are positively correlated at high-resolution with wetland
location leading to increasing $CH_4$ emissions with increasing resolution. Keeping track of wetland fraction reduces the
impact of resolution. However, uncertainties in $CH_4$ emissions remain high because of the large uncertainty in the
representation of wetland area, as demonstrated using output of the WetChimp intercomparison over our study domain.
Because of wetlands mapping uncertainties, the existing models are unlikely to realistically represent the correlation
between soil moisture and soil carbon availability. The correlation is positive in our simplified model, but may be different
in more complex models depending on their method of representing substrate availability. Therefore, depending on the
correlation, $CH_4$ emissions may be over or underestimated. As increasing the model resolution is an effective approach to
mitigate the problem of accounting for the correlation between soil moisture and soil carbon and improve the accuracy of
models, the main message of this study is that increasing the resolution of global wetland models, and especially the input
datasets that are used, should receive high priority.

## 1. Introduction

Despite decades of research, the main drivers of variations in the growth rate of atmospheric methane ($CH_4$) are still poorly
understood (Saunois et al., 2020). This is a critical knowledge gap, since $CH_4$ is the second most important anthropogenic
greenhouse gas after carbon dioxide ($CO_2$) (IPCC, 2013), and the increase of its recent growth rate introduces significant
uncertainty in the scenarios that are used in climate projections (IPCC, 2007; Bloom *et al.*, 2017). While those projections
are mainly concerned with anthropogenic emissions, natural emissions of $CH_4$ are important also since they account for an
important fraction of the growth rate uncertainty (Saunois et al., 2020; Bloom et al., 2016). This can be explained by the



poorly quantified response of these emissions to changing climatological conditions on a wide range of temporal and spatial scales (Bloom et al., 2016).

Natural wetlands have by far the largest contribution to the natural budget of methane estimated at 145 Tg $CH_4$ $yr^{-1}$, accounting for 20-40% of the total global $CH_4$ emission (Saunois et al., 2020). Generally, natural wetlands are defined as ecosystems with intermittent or permanent water saturated soils, such as peatlands (bogs and fens), mineral soil wetlands (swamps and marshes), determining the vegetation composition, productivity and nutrient cycling (Saunois et al., 2020). Due to their nature, wetlands are carbon-rich and moist environments favorable to microbes metabolizing organic matter

under anaerobic conditions leading to $CH_4$ production (Silvey et al., 2012). Estimates of $CH_4$ emissions from wetlands using a range of top-down and bottom-up techniques show a large inconsistency between the two approaches (Saunois et al., 2020) resulting from large uncertainties in the distribution and the underlying processes controlling the balance between microbial production and oxidation of methane (Kirschke et al., 2013; Melton et al., 2013a).

Models used for quantifying $CH_4$ emissions vary in their methodology and level of detail. The WetChimp model inter-
comparison (Melton et al., 2013b) highlighted the variety of models that are used, and the wide range of global and regional emissions resulting from them. The global distribution and area of wetlands, determined either using wetland maps, hydrological modelling, or satellite-derived inundation maps, was identified as a main source of uncertainty (Melton et al., 2013a; Wania et al., 2013). It was concluded that the simulated wetland extents are difficult to evaluate due to extensive disagreements between remotely sensed inundation datasets and wetland mapping (Wania et al., 2013). The simulated

global wetlands $CH_4$ flux estimates were in the range of 141 to 264 Tg $CH_4$ $yr^{-1}$ with a mean value of 190 Tg $CH_4$ $yr^{-1}$ (Wania et al., 2013), which is in the upper part of the large uncertainty range of the early estimate by Matthews and Fung (1987) of 10 to 300 Tg $CH_4$ $yr^{-1}$. Despite the progress in narrowing that range, the uncertainty in global wetland $CH_4$ emissions remains very high (Wania et al., 2013).

The recent global $CH_4$ budget study of Saunois et al. (2020) compared 13 models for the 2008-2017 period, resulting in
somewhat lower global emissions in the range of 101 and 179 Tg $CH_4$ $yr^{-1}$ with an average of 148 Tg $CH_4$ $yr^{-1}$. Again, it was concluded that wetland extent appears to be the main contributor to uncertainties in the absolute flux of $CH_4$ emissions from wetlands, as well as in other recent studies on this topic (Zhang et al., 2017b; Anthony Bloom et al., 2017; Peltola et al., 2019). Zhang et al. (2017) concluded that it is an important need for the scientific community to construct a global scale wetland dataset at high spatial and temporal resolution, by integrating multiple sources of field and satellite data using

models.

Zhu et al. (2014) investigated the sensitivity of $CH_4$ emisions from pan-arctic wetlands to the spatial resolution of water table depth, comparing simulations at 5 and 100 km resolution. The significant differences that were found, suggest that macro-scale biogeochemical models using grid-cell-mean water table depth might have underestimated the regional $CH_4$ emissions. It was recommended to consider the spatial scale-dependence of $CH_4$ emissions on water table depth in future

quantifications. Awuah (2017) examined the influence of spatial resolution and land-cover heterogeneity on the accuracy of land cover mapping. It was concluded that spatial resolution plays an important role in classifying land cover. The fraction of mixed ecosystem pixels decreased and the overall classification accuracy improved when the spatial resolution was increased.





In this study, we investigate the importance of spatial resolution for the quantification of methane emissions from natural
wetlands, and whether the use of high-resolution wetland maps may be an effective strategy for reducing its uncertainty.
The sensitivity of emissions to spatial resolution is tested for the Fennoscandinavian domain, using a high-resolution
wetland map ($100 \times 100 m^2$) in combination with a highly simplified $CH_4$ emission model. The advantage of using a simple
model is its numerical efficiency as well as the conceptual ease to control and understand the relations that are found. The
wetland map is coarsened across a wide range of resolutions, including those that are commonly used in global wetland
models. Site measurements from Finland are used to calibrate the model, before it is used to compute the impact of
resolution on the integrated $CH_4$ emission over the study area. Finally, wetland extent maps from the WetChimp models
intercomparison inventory are used to assess the significance and realism of the results obtained using the simple model.

Four sections follow this introduction. Section 2 presents the methodology used in the study, introducing the model, study
area, and data sets that are used. Section 3 presents the results quantifying the resolution dependence in the simple model.
Section 4 discusses the interpretation and significance of the results, in light of its limitations, and in comparison with the
WetChimp models. Finally, Section 5 presents the main conclusions, the most critical remaining uncertainties, as well as
recommendations for future research.

## 2. Methodology

### 2.1 Hypothetical Experiment

The principle of the resolution-dependence we investigate can be explained using a simple hypothetical case. Let us assume
a wetland area W that can either be described at high-resolution by 2x2 tiles, each with area $A_T=1$, or at low resolution by
combining the 2x2 tiles into a single tile of $A_T=4$ (see Figure 1). To quantify the $CH_4$ emission in these tiles, we use the
highly simplified model of wetland emissions,

$$FCH_4 = SC.SM.A_T \qquad (1)$$

in which the $CH_4$ emission ($FCH_4$) is the product of the availability of soil carbon (SC), soil moisture (SM), and the area
of each wetland tile ($A_T$). This model is a simplified version of $CH_4$ emission equation that we will use in the remainder
of this study, as will be explained in section 2.2.

In the first case, we set SC and SM both to unity. As a result, in the high-resolution representation (Figure 1.a) each cell
has an emission of 1, and therefore the total emission over wetland area W equals to 4 (in an arbitrary unit). When
aggregating the high-resolution tiles to coarse resolution (Figure 1.b), soil moisture and soil carbon are the average of the
2x2 tiles. In this case, the sum of emissions will again be 4, i.e. the high- and low-resolution representations are consistent.





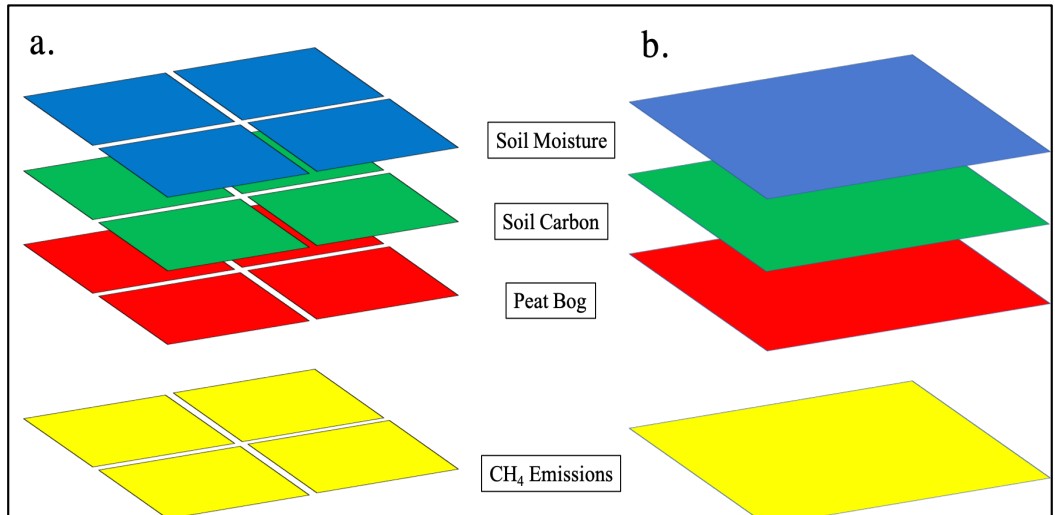

**Figure 1: A hypothetical case of wetland CH₄ emissions, represented at high-resolution (a) and low resolution (b). The CH₄ emission is calculated using input data fields of soil moisture, soil carbon and a wetlands mask, each at the same resolution.**

Alternatively, we assume that wetlands are located in two cells only (Figure 2.a), the other two cells being upland. For upland tiles we assume soil carbon and soil moisture to be zero, so that wetland CH₄ emissions are zero also. Then when applying the same principle, the total emission in the high-resolution case is 2 (Figure 2.a), whereas it is 1 (4x0.5x0.5) in the low-resolution case (Figure 2.b). Here the resolution-dependence of the CH₄ emission arises because of the product SC x SM in Eq. (1), causing the impact of the averaging to coarse resolution on the CH₄ emission to be squared.

This outcome can be generalized to larger steps in resolution as follows:

$$E_{LR} = \overline{SC}.\overline{SM}.A_{LR} = (n_{wl}SC_{wl}A_{HR}/A_{LR})(n_{wl}SM_{wl}A_{HR}/A_{LR})A_{LR} \qquad (2)$$

$$E_{HR} = n_{wl}SC_{wl}SM_{wl}A_{HR} \qquad (3)$$

$$\frac{E_{HR}}{E_{LR}} = \frac{n_{wl}SC_{wl}SM_{wl}A_{HR}}{(n_{wl}SC_{wl}A_{HR}/A_{LR})(n_{wl}SM_{wl}A_{HR}/A_{LR})A_{LR}} = \frac{A_{LR}}{n_{wl}A_{HR}} = \frac{1}{F_{wl}} \qquad (4)$$

where $E_{HR}$ and $E_{LR}$ are the emissions of the coarse resolution grid box evaluated at respectively high and low resolution.

$A_{HR}$ and $A_{LR}$ are the grid box areas at high and low resolution, and $n_{wl}$ is the number of high-resolution grid boxes that are covered by wetland (note the use of the subscript $wl$ to indicate a wetland grid box). If $F_{wl}$ is the wetland fraction, then the right-hand-side term in equation 4 is $1/F_{wl}$. As long as the wetland fraction remains the same, the impact of a change in resolution will remain the same also. However, if the wetland fraction becomes lower, because part of coarse resolution wetland grid box happens to be dry at high resolution, then the impact of a change in resolution increases.




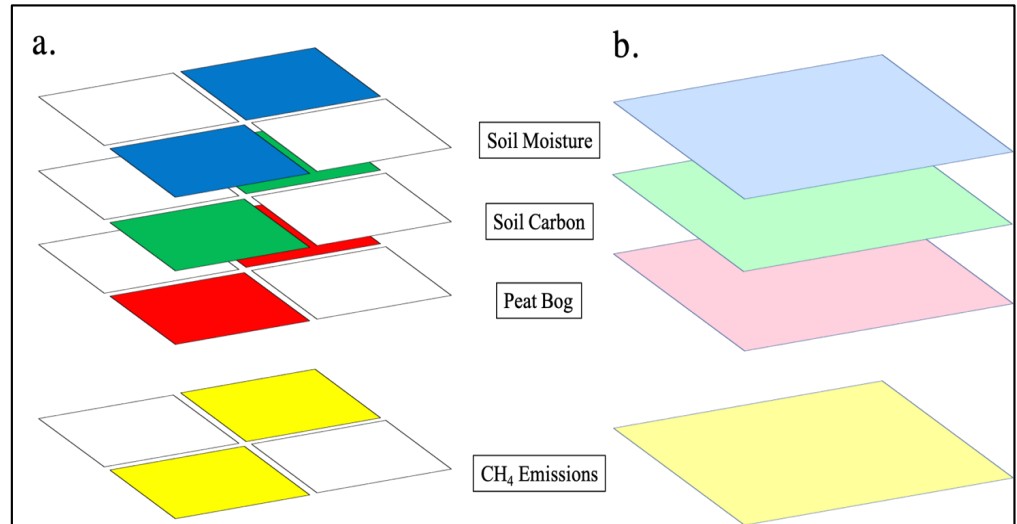

**Figure 2: As Figure 1 for a case in which wetlands occupy 50% of the total area.**

## 2.2 Wetlands CH₄ Model

The CH₄ emission scheme of Gedney et al. (2004) is used to compute CH₄ emissions for the case study area described in

section 3. It is a highly simplified representation of wetland CH₄ emissions, but well suited for testing resolution dependences because of its computational efficiency and ease of interpretation as the number of model parameters is only small. The CH₄ flux from wetlands $F_{CH4}$ [g CH₄ m$^{-2}$ yr$^{-1}$] is calculated from the basic CH₄ controls of soil temperature ($T_{soil}$), soil moisture (SM) and soil carbon (SC), as follows:

$$F_{CH4} = K_{CH4}.SC.SM.Q_{10}^{(Tsoil-T_0)/10} \qquad (5)$$

Where $T_{soil}$ is the average soil surface temperature in Kelvin [K] for the top 5 cm. $Q_{10}$ is the temperature sensitivity of the CH₄ emission to a 10 K temperature change relative to $T_0 = 273.15$ K. Since $F_{CH4}$ is now expressed as the CH₄ flux per unit area, this will be used for soil carbon also (SC in [g.m$^{-2}$]). $K_{CH4}$ is a calibration constant relating the driving variables to a CH₄ flux in units of [g CH₄ m$^{-2}$ yr$^{-1}$]. We want to note here that the input data used in Eq.5 are for year 2015 as will be described in section 3.2.

Different scenarios are used (Sn.1 – Sn.4) representing wetlands, uplands, and combinations between them (Table 1). In Sn.1, we use the high-resolution wetlands map (see section 3.2) as a mask for wetlands to distinguish wetlands from the upland surroundings. CH₄ emissions are only calculated for the wetland fraction $F_{wl}$. This is because equation 5 does not apply to aerobic upland soils, where CH₄ oxidation by methanotrophic bacteria dominates methanogenic CH₄ production.

Despite that, in Sn.2 uplands are treated as the wetlands in Sn.1. CH₄ oxidation in upland soils may show a resolution-

dependence following the logic of section 2.1 also. However, since the upland fraction is generally substantially larger than the wetland fraction at spatial resolutions that are common in global wetland modelling, the sensitivity of the sink to resolution is expected to be less important (see equation 4). The setup of Sn.2 is meant to isolate the impact of the difference between wetland and upland fraction on the resolution dependence, which explains why the method to compute the flux is



kept the same. Sn.3 combines wetlands and uplands to test the impact of changing the contrast between uplands and
wetlands emissions, using the same threshold values of SC and SM in Sn.1 and Sn.2. Sn.4 represents emissions from
wetlands only, like in Sn.1, but using spatially varying SM and SC data (section 3.2.). The aim is to test the extent to which
the results of Sn.1 and Sn.2 might have been influenced by the simplifying assumptions on SC and SM that are made, and
how sensitive the resolution-dependence may be to a more realistic representation of their spatial variations.

The first three scenarios used to test the resolution-dependence were aggregated from original high-resolution wetlands
datasets described in the data section to 6 different resolutions; 0.005º, 0.01º, 0.05º, 0.1º, 0.5º and 1º. For the remained
scenario, we aggregate from the finest available resolution of the global hydrological model PCR-GLOBWB (PCRG) (5
arcmin) to 0.1º, 0.5º and 1º.

We acknowledge that our wetland 'model' provides only a highly simplified representation of the processes controlling
$CH_4$ emissions in wetlands. However, the main objective is to demonstrate the principle and provide a first order estimate
of its importance, suitable to provide a basic discussion to be refined further using more sophisticated models in the future.

**Table 1: List of Scenarios**

| Scenarios | Wetlands | | Uplands | | Temperature [K] |
|---|---|---|---|---|---|
| | SC [g.m$^{-2}$] | SM [cm$^3$.cm$^{-3}$] | SC [g.m$^{-2}$] | SM [cm$^3$.cm$^{-3}$] | |
| Sn.1 | 110 | 0.70 | 0 | 0 | ERA-5 |
| Sn.2 | 0 | 0 | 10 | 0.10 | ERA-5 |
| Sn.3 | 110 | 0.70 | 10 | 0.10 | ERA-5 |
| Sn.4 | ISRIC2017 | PCRG | 0 | 0 | ERA-5 |

## 3.    Study area and Data

### 3.1    Study area

The Fennoscandinavian peninsula, excluding the Russian sector, is used as the domain of our computations (see Figure 3).
It is chosen as a favorable compromise between size, importance for high-latitude $CH_4$ emissions, ecosystems diversity,
and data availability. In this domain, $CH_4$ fluxes are monitored at a few sites that are reporting to FLUXNET-CH4
(https://fluxnet.org). Despite the limited number of sites (2 sites in this study), the network density is still relatively high
for the circumpolar boreal/arctic region.





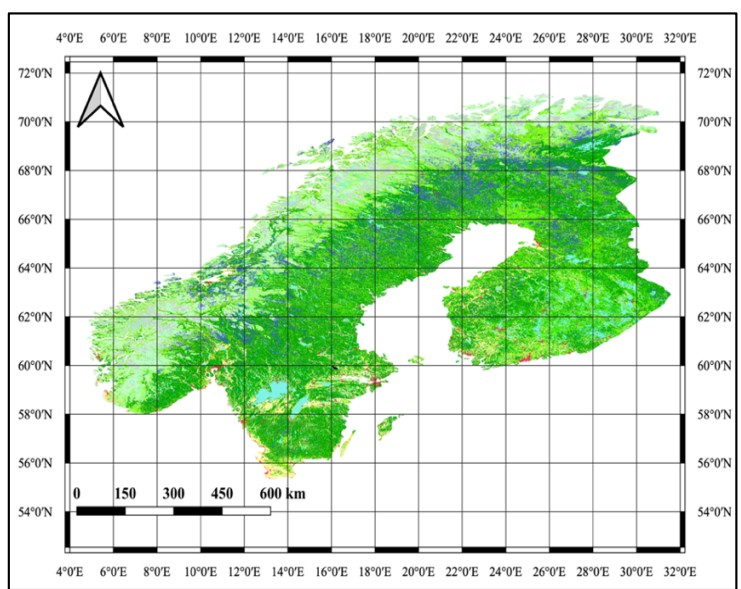

**Figure 3: Study area domain and land cover classification (for the color legend see Figure A-3).**

### 3.2 Data

**Wetland map**

To localize wetlands at high resolution, the Corine Land Cover map is used (CLC2018). These data are made available by the Copernicus Land Monitoring Service from (https://land.copernicus.eu/pan-european/corine-land-cover/clc2018).

CLC2018 represents the year 2018 at 100x100m$^2$ resolution for the European continent (Büttner et al., 2017), and provides information about the physical state of the landscape (Faltan et al., 2020). The land-cover classification is based on satellite images with a spatial resolution in the order of meters, from sensors onboard Landsat, RapidEye, Sentinel-2, and Landsat-8. This information is extended using various auxiliary data, e.g., aerial photographs, thematic maps, etc. yielding a high resolution land cover map suitable for large scale research and land cover/use mapping (Faltan et al., 2020). CLC2018

classifies wetlands into major categories; inland wetlands and coastal wetlands. Inland wetlands are inland marches and peatbogs (class number 411 and 412 respectively), which we use in our study as decribed in the CLC2018 user guide (Kosztra et al., 2017).

**Organic Soil Carbon SC**

To specify SC in equation 2, the soil carbon dataset from the International Soil Reference and Information Centre (Hengl

et al., 2017) at 250x250m$^2$ resolution is used. According to the ISRIC map, the SC for the study area ranges between (10-110 g.m$^{-2}$). This information is used in different ways as specified in Table 2. In scenario 4 and 5 (Sn.4 & Sn.5), the full soil carbon map is used. In Sn.1 and Sn.3, however, we use the maximum value in the ISRIC map to represent peat. The underlying assumption is that soil carbon in the ISRIC map is limited by the peat fraction at 250x250m resolution, and that the highest values represent grid boxes that are fully covered by peat. In Sn.2 the lowest value is used to represent uplands



conditions, following the same logic. Note that this is a simplifying assumption since different land cover types have different soil carbon contents, but this choice guarantees the expected insignificant contribution of methane emissions from upland ecosystems. The ISRIC data were downloaded from (https://files.isric.org/soilgrids/former/2017-03-10/data ).

**Soil Moisture SM**

The study of Schaufler *et al.* (2010) hypothesized that $CH_4$ fluxes peak at soil moistures between 30% and 70% of water-
filled pore space and decline below 20% and above 80%. Following this hypothesis, for Sn.1, Sn.2 and Sn.3 we minimize SM for uplands to be 0.10 $cm^3.cm^{-3}$ again as an attempt to lower the impact of upland resolution-dependence when following the coarsening steps explained in the following sub-section. SM is maximized at wetlands locations to 0.70 $cm^3.cm^{-3}$.

For Sn.4, we use soil moisture data modelled by PCRG. PCRG is grid-based global hydrology and water resources model
(Sutanudjaja et al., 2018). We run the model to simulate soil moisture for the study area using the two available versions with spatial resolutions of 5 arcmin (≈10km) and 30 arcmin (≈50km). PCRG has three different soil depth layer, 0-5 cm (top layer), 5-30 cm and 30-150 cm. A simulation has been carried out for the year 2015 and output has been used for the top 5 cm soil layer, after remapping the data to 0.1º and 0.5º resolution.

**Temperature T**

Temperature in equation 5 has been taken from the European Centre for Medium-range Weather Forecasts reanalysis (ECMWF-ERA5), using daily soil surface temperature for the top 5 cm of the soil for the year 2015. Data at a spatial resolution of 7x7 $km^2$ were downloaded from (https://www.ecmwf.int/en/forecasts/datasets/reanalysis-datasets/era5). We acknowledge that this resolution should preferably have been higher. Although air temperature variations may be represented adequately enough at this resolution, the surface energy balance of wetlands and upland ecosystems is expected
to be different. This gives rise to variations in soil surface temperature that we are unable to account for but are assumed to be second order in importance compared to variations in soil carbon and soil moisture.

**$Q_{10}$ and $K_{CH4}$**

For the temperature sensitivity of methane emissions from natural wetlands, previous studies derived $Q_{10}$ values varying in the range of 1.7–16 (Walter and Heimann, 2000). This wide range is explained by the difficulty of separating co-varying
environmental drivers (Gedney et al., 2004). We use a $Q_{10}$ of 3.0, following the studies of Wania *et al.*, (2013), used also in the WetChimp simulations (Melton *et al.*, 2013). Ringeval *et al.*, (2010) derived this value in an attempt to optimize the agreement between the LPX model and site measurements under inundated conditions.

To estimate the $K_{CH4}$ emission calibration factor, we use daily $CH_4$ flux measurements for the year 2015 from sites located within the study area (Table 1). The $K_{CH4}$ that is used brings our simulations approximately to the same annual emission
for the year 2015 as measured at the FLUXNET sites Siikaneva, Finland, and Degero, Sweden (see section 4.4). $CH_4$ flux measurements have been downloaded from (https://fluxnet.org/download-data/).

**Table 2: Measurements Sites Used for Calibrating $K_{CH4}$.**

| Site ID | Site Name | Country | Lat (°N) | Lon (°E) |
|---------|-----------|---------|----------|----------|
| FI-Siik | Siikaneva I | Finland | 61.83 | 24.19 |
| SE-Deg | Degero | Sweden | 64.18 | 19.56 |



## 4. Results

This section presents the modelled $CH_4$ emissions over our Fenno-Scandinavian domain for the scenarios in Table 2, using the datasets described in section 3.2, and how they vary with spatial resolution. Annual $CH_4$ emissions integrated over the full domain span a wide range when coarsened from the highest resolution of 0.001°, used as reference, to progressively coarser resolutions up to 1° (Tables B-1 and B-2).

### 4.1 Scenario Sn.1

Figure 4 compares the spatial distribution of annual $CH_4$ emissions from wetlands over the study area. Significant differences are seen across the wide range of scales from the reference resolution to the coarsest resolution of 1°x1°. Towards the finer resolutions, the spatial pattern gradually converges, however, the reference resolution integrated $CH_4$ emissions is ~1.68 Tg $CH_4$ yr$^{-1}$ which about two times higher than at 0.005° resolution (~0.96 Tg $CH_4$ yr$^{-1}$). The total emission difference increases aggregating to coarser resolutions. The total emissions decrease from 1.68 Tg $CH_4$ yr$^{-1}$ at the finest resolution to ~0.13 Tg $CH_4$ yr$^{-1}$ at the coarsest resolution of 1°x1°. To eliminate coastal area effects on the resolution dependence, due to coarsening the land sea mask, we exclude all cells nearby the shoreline such that grid boxes at the coarsest resolution are still entirely over land. This takes care in the same way for the border with Russia.

**Table 3: Integrated $CH_4$ emissions for the study area (All scenarios).**

| Integrated methane emissions over the study area | | | | | | | | |
|---|---|---|---|---|---|---|---|---|
| **Scenarios** | Resolution [°] | **0.001** | **0.005** | **0.01** | **0.05** | **0.1** | **0.5** | **1** |
| **Sn.1** | Total Emissions [Tg $CH_4$ yr$^{-1}$] | 1.68 | 0.966 | 0.789 | 0.335 | 0.251 | 0.155 | 0.131 |
| | $CH_4$ [Ref. Resolution/Resolution#] | **1** | **1.73** | **2.33** | **4.99** | **6.89** | **10.85** | **12.86** |
| **Sn.2** | Total Emissions [Tg $CH_4$ yr$^{-1}$] | 0.12 | 0.1182 | 0.1182 | 0.1179 | 0.1177 | 0.1165 | 0.1152 |
| | $CH_4$ [Ref. Resolution/Resolution#] | **1.00** | **1.02** | **1.02** | **1.02** | **1.02** | **1.03** | **1.04** |
| **Sn.3** | Total Emissions [Tg $CH_4$ yr$^{-1}$] | 1.8 | 1.0842 | 0.9072 | 0.4529 | 0.3687 | 0.2715 | 0.2452 |
| | $CH_4$ [Ref. Resolution/Resolution#] | **1.00** | **1.66** | **1.98** | **3.97** | **4.88** | **6.63** | **7.34** |

Ref. Resolution is the emission at the highest resolution (0.001°)

Resolution# is the emission from tested resolutions 0.005°,0.01°,0.05°,0.1°,0.5° and 1°.



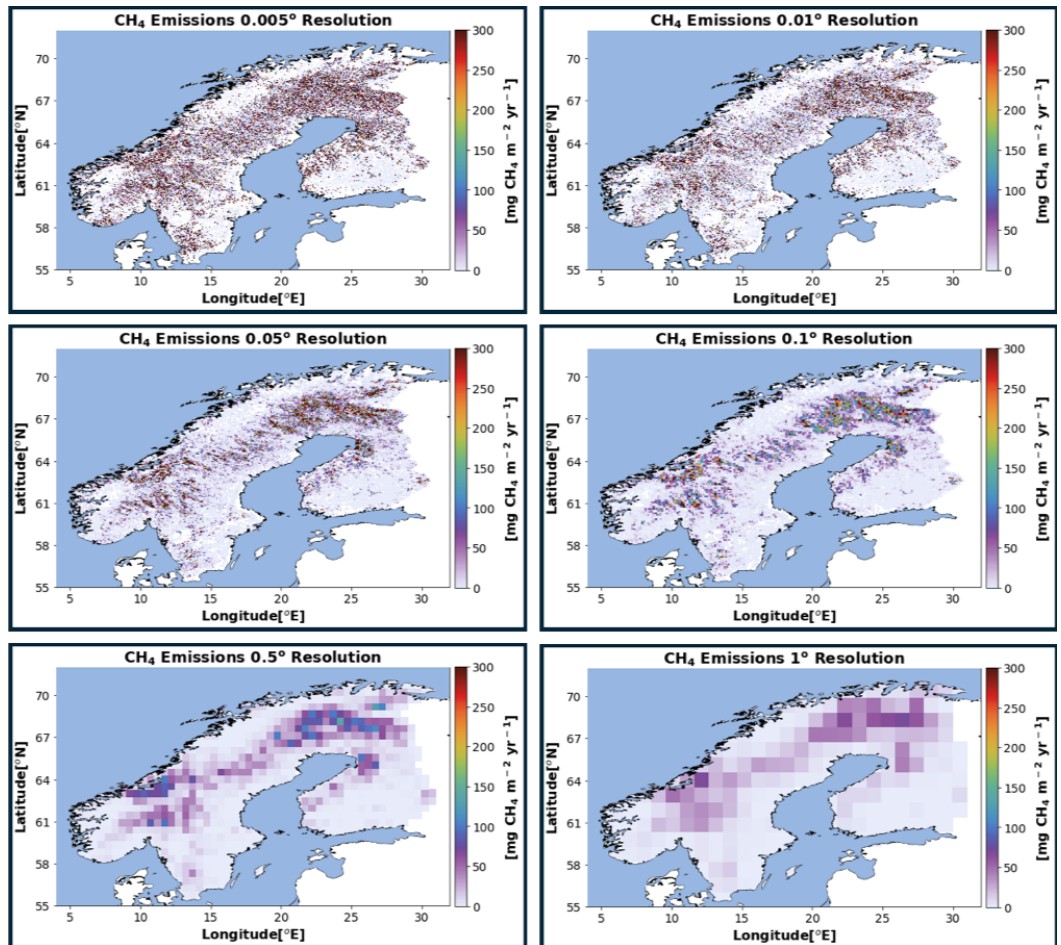

**Figure 4: CH₄ emissions of Sn. 1, spanning the full range of resolutions from 0.005º (top left) to 1º (bottom right).**

### 4.2 Scenario Sn.2

The resolution dependence for CH₄ emissions (uptake in reality) in uplands soils is less strong than for wetlands. Figure A.1 compares total CH₄ emission for the study area obtained using prescribed values for SC and SM in Table 2. The emission ratio of each test resolution remains much closer to unity compared to Sn.1 (Table B.1). The total emission at the uplands reference-resolution (0.12 Tg CH₄ yr⁻¹) is only a factor of 1.04 higher than at the coarsest resolution of 1ºx1º (0.115 Tg CH₄ yr⁻¹). Since the study area is dominated by uplands, the impact of averaging to coarser resolution is expected to be less than for wetlands.

### 4.3 Scenario Sn.3

In this scenario, we combine Sn.1 and Sn.2 (i.e. wetlands and uplands) to account for the fact that SC and SM are both larger than zero for uplands. Accounting for the availability of soil moisture and soil carbon in upland soils reduces the difference between upland and wetlands values, which is expected to reduce the resolution dependence. Note that in this scenario, the results assume that emissions are always positive in





upland ecosystems, which usually is not the case. Nevertheless, the reason for including this scenario is to test the effect of assuming larger upland/peatland contrasts in SC and SM than will be the case in reality. As can be seen the resolution effect in this scenario is less than in Sn.1 but still large.

In Figure 5, scenarios Sn.1 – Sn.3 have been plotted together to show the difference in resolution dependence between them.

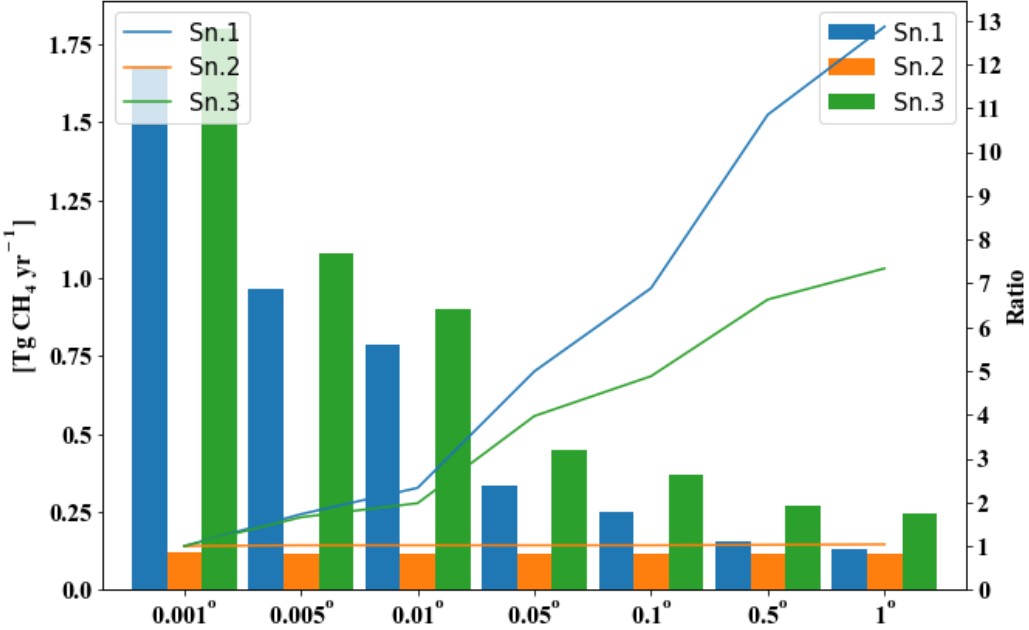

**Figure 5: Resolutions dependence for Sn.1 – Sn.3.**

### 4.4    Scenario Sn.4

Here, the resolution dependence is computed using daily varying SC, SM and $T_{soil}$ data (see Table 2). Following the same steps, using the finest resolution PCRG soil moisture data (5 arcmin). The latter is regridded to 0.1º(~10x10km) from which coarsened maps at 0.5º(~50x50km) and 1º(~100x100km) are derived by aggregation. To further improve the realism of the simulation, the emission model at 0.1º resolution has been calibrated using flux measurements. This is done in two ways; first, as the $K_{CH4}$ values (Figure 6) are different for both Degero and Siikaneva sites at 0.1º, the $K_{CH4}$ for both sites were

averaged (value = 0.025) and applied to Eq.5 to simulate CH4 at 0.1 º ,0.5º and 1º as well as the simulated emissions in scenarios 1-3. The results show considerable differences between the three modelled resolutions. At 0.1º resolution, we find an integrated CH4 emissions from wetlands of ~ 0.021 Tg yr$^{-1}$, which reduces by ~14% for the 0.5º to ~ 0.018 Tg yr$^{-1}$ and reduces by ~24% for the 1º to ~ 0.016 Tg yr$^{-1}$ (Figure 7).

On the other hand, if we calibrate the results of each resolution used in this scenario with site measurements, so that each

modelled resolution agrees with the measured annual total, this results in different $K_{CH4}$ values for each of the tested resolutions (Figure 6). $K_{CH4}$ values at Degero and Siikaneva decrease with resolution, but not by much (about 10%). Using these $K_{CH4}$ values, we find a domain integrated CH4 emissions from wetlands of ~0.021 Tg yr$^{-1}$ at 0.1º resolution, which reduces by ~10% for the 0.5º to ~0.019 Tg yr$^{-1}$ and reduces by ~19% for the 1º to ~0.017 Tg yr$^{-1}$. This means that the use





of different $K_{CH4}$ values partially mitigates the resolution dependence, but not enough to fully account for it. Note that this

result will depend, among different factors, on the size of the wetland for which measurements are available. However, for wetlands that are smaller than the coarsest resolution grid box, the impact is expected to be in the direction that we find for Siikaneva and Degero. This result favors the use of high-resolution models, for which the calibration will be most accurate. However, it argues against to use of high-resolution $K_{CH4}$ values in coarser resolution models.

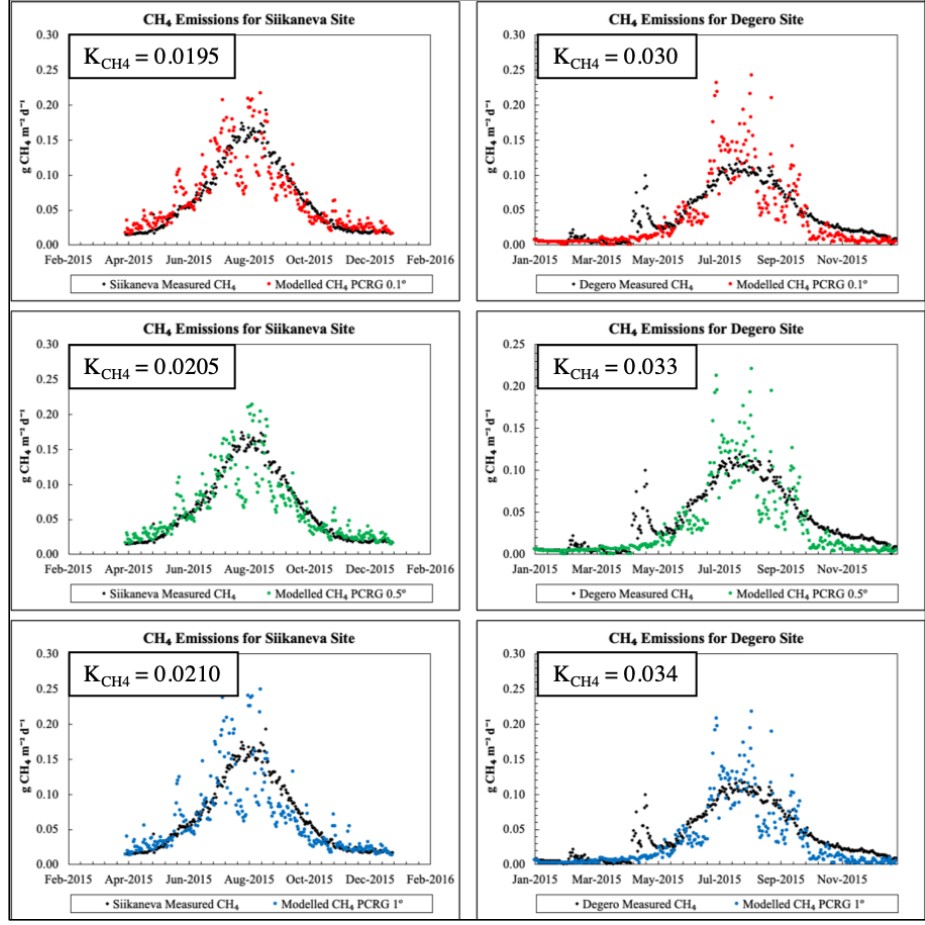

**Figure 6: Siikaneva-Finland (left) and Degero-Sweden (right) CH$_4$ Observations and Calibrated Model Estimates at the 0.1°, 0.5° and 1° resolution.**


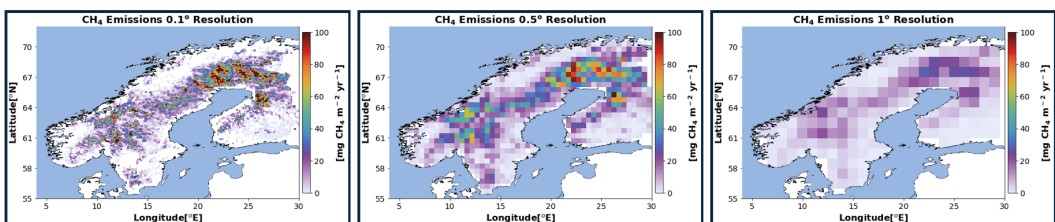

**Figure 7: Integrated CH$_4$ emissions for wetlands over the study area using PCR-GLOBWB soil moisture inputs at 0.1° (left) and 0.5° (middle) and 1° (right) for Sn.4.**





## 5. Discussion

The results of our simplified wetland experiments show a strong dependence of regionally integrated $CH_4$ emissions on the spatial resolution that is used. The question, however, is whether this resolution dependence is representative of wetlands models that are used to estimate wetland $CH_4$ emissions or that it arises because of the simplicity of the setup that has been chosen. One obvious simplification is the use of grid box averaged soil carbon and soil moisture values. Wetland models commonly keep track of the sub grid fraction that is covered by wetlands. In our simplified experiment, that approach fully accounts for the resolution dependence. This can readily be understood from Eq. (4), indicating that the resolution dependence scales with the inverse wetland fraction (the right-hand-side being $1/F_{wl}$). Therefore, if the soil carbon and soil moisture are averaged over the wetland fraction rather than the whole grid box, then the $E_{HL}/E_{LR}$ ratio becomes 1 and the resolution dependence vanishes.

However, a few problems remain. The first is that the wetland fraction is determined from a hydrological model or satellite data with a limited horizontal resolution, compromising the ability to determine the wetland fraction. Secondly, the representation of wetland area in models is associated with large uncertainties.

To assess the uncertainty in wetland area, we have plotted the wetlands extent maps used by the WetChimp model intercomparison (Melton et al. 2013) (Table B. 3) for the Fennoscandinavian Peninsula in Figure 9. For reference, the high-resolution CLC2018 wetland map is included at the same resolution of $0.5^{\circ} \times 0.5^{\circ}$ to match with the resolution of wetlands maps of WetChimp. Depending on the type of information that is used to determine where the wetlands are, the wetland map looks very different. Integrated over our domain, the total wetland areas represented by the models (Figure 10) are significantly different and range between $53 \times 10^3$ and $171 \times 10^3$ $km^2$. The Swedish Wetland Survey (VMI) reports a total wetland area of approximately $34 \times 10^3$ $km^2$ for Sweden (Gunnarsson and Löfroth, 2014). According to Ramsar, however, the Swedish wetland areal extent amounts only to 6655 $km^2$. If the VMI estimate from Sweden is combined with Ramsar estimates for Finland and Norway of 7795 $km^2$ and 9091 $km^2$ (Ramsar, 2021) respectively, this leads to a total wetland area for the Fennoscandinavian peninsula of $51 \times 10^3$ $km^2$. which is in close agreement with the Corine land cover map ($53 \times 10^3$ $km^2$).

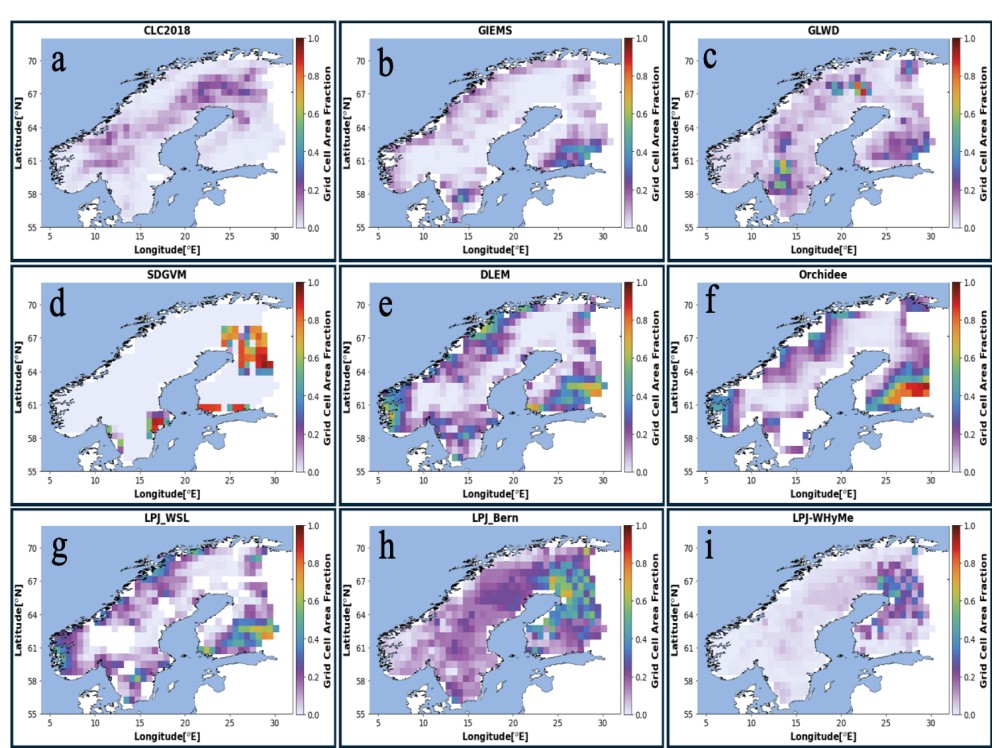

Figure 8: Wetland extent maps used by the participated models of WetChimp intercomparison (from b to i) in comparison to
CLC2018 wetland extent map (a).

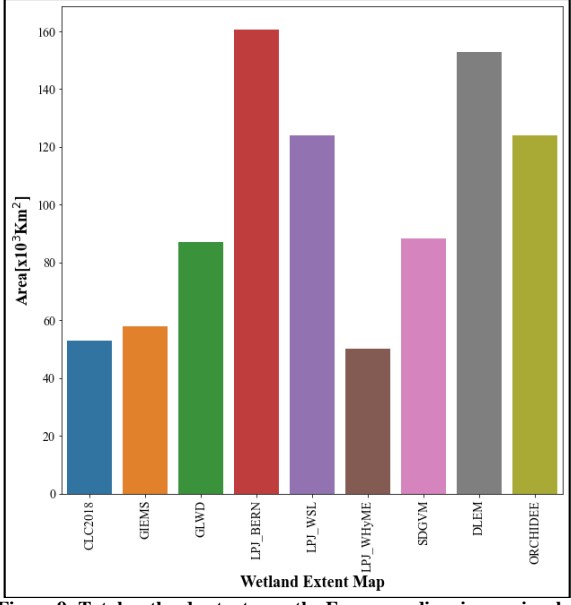

Figure 9: Total wetland extent over the Fennoscandinavian peninsula.





Looking at the overall pattern of modelled wetland extent, most of the models simulate greater wetland area than CLC2018

(Figure 10). LPJ-WHyMe is in closest agreement with CLC2018 for the total wetland area (see Figure 10). However, its
spatial distribution of wetlands is very different. The maps in Figure 9 and corresponding correlation matrix in Figure 11,
show large disagreements in magnitude and spatial distribution of wetland extent among the WetChimp datasets. This is
primarily due to inconsistencies in 1) the definition and classification of wetland types (e.g. peatland or inundated area), 2)
the time window represented by the wetland datasets, 3) the purpose of the wetland data set and the method from which it

was derived  (Zhang et al., 2017a).

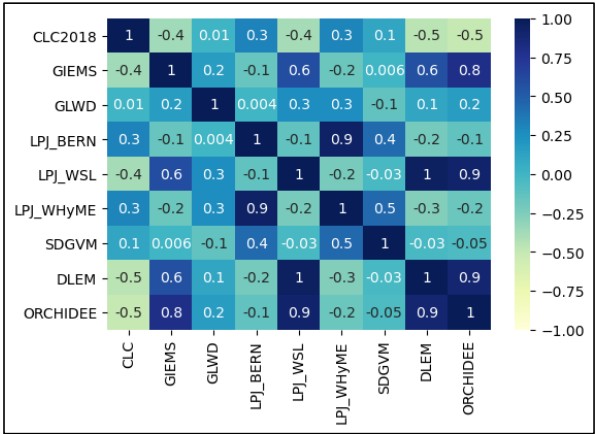

**Figure 10: Correlation matrix for the tested wetlands datasets used in the study.**

The importance of uncertainties in wetland area have been reported before (Wania et al., 2013). The reason for mentioning
it here is the implication for the correlation between wetland location and other variables, such as soil carbon and soil

moisture, which are multiplied to compute $CH_4$ emissions as in Eq. (2). It is the correlation between these terms that
determines the resolution dependence. To show this, we simplify Eq. (2) further so that only variations in soil moisture and
soil carbon are considered. In this case, Eq. (4) can be reformulated, expressing local soil moisture and soil carbon as sums
of their coarse resolution mean ($\overline{SM}, \overline{SC}$) and the local deviation ($\Delta SM[i], \Delta SC[i]$). This leads to

$$\frac{E_{HR}}{E_{LR}} = 1 + \frac{1}{\overline{SC}\,\overline{SM}\,A_{LR}} \sum_{i=0}^{i=n} \Delta SC[i] \; \Delta SM[i] \; A_{HR} \tag{6}$$

with *n* the number of high-resolution grid boxes in each low-resolution grid box. Eq. (6) shows that for uncorrelated soil
carbon and soil moisture, the second right-hand-side term becomes small and the ratio approaches 1. For a positive
correlation, the emission increases with resolution. The effect is large if local deviations are large compared with the coarse
resolution mean. Likewise, negative correlations lead to emissions that decrease with increasing resolution. This equation

explains why emission scenarios with smaller differences between upland and wetland soil carbon and soil moisture lead
to smaller resolution dependences. To avoid resolution dependent errors, it is important to get the correlation between soil
carbon and soil moisture right. The same is true for temperature variations, following the same logic. The challenge of
getting the spatial correlation right is highlighted in Figure 11, which shows the limited correlation (-0.12 on average) in
wetland area between the WetChimp models over Fennyscandinavia.



Because of Eq. (6), the use of wetland fractions is only sufficient to deal with resolution dependence if there is no variation between sub grid wetland regions – the opposite is generally the case for wetlands, as their $CH_4$ emission is known to be highly heterogeneous. We have tried to quantify the resolution dependence that might arise from variations within the wetland fraction. The results (not shown) indicate that the impact is only small (2%). However, it is questionable how well the ISRIC and PCR-GLOBWB (corr. = 0.89 at 0.1° resolution) datasets capture the variability at their native resolutions.

The role of soil carbon requires special attention, because many models rather use soil respiration or vegetation productivity as measure of the amount of available degradable carbon. However, here no distinction is made between wetland and upland productivity, whereas in reality the productivity in wetlands is usually much lower than in uplands due to oxygen limitation. As a result, important errors are to be expected from models failing to capture the correlation of the parameters that drive $CH_4$ emissions from wetlands.

A solution to mitigate resolution dependent errors is to increase resolution up to Eddy Covariance tower (EC) resolution, which is 100x100m in order to calibrate model results to EC measurements. As shown in this study, advanced datasets are available for doing this. Equally important to get the correlation right is it for these datasets to be mutually consistent. Note that this is true not only for the distinction between wetland and upland ecosystems. Large variations occur also within a single wetland.

Multivariant Probability Density Functions (PFDs) might be useful to mitigate this problem by determining the correlation between SM and SC at high resolution maps then identifying the multi-variant PDF of SM and SC at the course resolution. We do not provide a solution for that, but argue that an important step in the right direction can be made using high-resolution datasets that are available.

**6. Conclusion**

This study investigates the dependence of regionally integrated $CH_4$ emissions on spatial resolution. Simulations are performed for the Fennoscandinavian domain at resolutions ranging from 100x100 $m^2$ to 1°x1°. The results of our simplified wetland experiments show that this dependence can be strong (up to 13 times greater between high "in meters" and coarse resolution). In the model that is used, the effect arises from the correlation between soil moisture and soil carbon. In our experiments, the impact is effectively mitigated by accounting for the sub grid wetland fraction. How well this works

dependents on how well the true wetland fraction is represented, which is a key uncertainty in wetland models. In addition, correlated variations between soil moisture and soil carbon within the wetland fraction lead to resolution dependent errors, which are more difficult to quantify using the available datasets. The results suggest that macroscale biogeochemical models might underestimate regional $CH_4$ emissions due to a coarse representation of the correlation between input parameters that drive the methane emission (such as soil moisture and soil carbon). Our solution is not a straightforward

recipe; however, we strongly recommend to make use as much as possible of existing high-resolution datasets.





*Data availability.* The data used in this paper are CLC2018 (https://land.copernicus.eu/pan-european/corine-land-cover/clc2018), ISRIC (https://files.isric.org/soilgrids/former/2017-03-10/data), ERA5 soil surface temperature (https://www.ecmwf.int/en/forecasts/datasets/reanalysis-datasets/era5). $CH_4$ flux measurements have been downloaded
from (https://fluxnet.org/download-data/). PCR-GLOBWB 2 is open source and distributed under the terms of the GNU General Public License version 3, or any later version, as published by the Free Software Foundation. The model code is provided through a GitHub repository: (https://github.com/UU-Hydro/PCR-GLOBWB_model).

*Author Contributions.* Y.A performed simulations, data analysis, interpretation and writing paper. SH supervised the study.
SH, and YV discussed the result. All the authors commented on the manuscript and improve it.

*Competing interests.* The authors declare that they have no conflict of interest.

*Acknowledgements.* We would like to thank PCR-GLOBWB team for the guidance and help to run the model. We also thank our reviewers for their constructive comments and thoughtful suggestions. We thank the PIs of the FLUXNET sites used in this study for making the datasets available to the research community. All the flux and meteorological data are
available from the FLUXNET2015 Dataset website (http://fluxnet.fluxdata.org/data/fluxnet2015-dataset/).

*Financial support.* The project is funded by the VU Amsterdam, under the carbon cycle data assimilation in the modelling of CH4 emissions from natural wetlands (project no. 2922502).



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





**Appendix A**

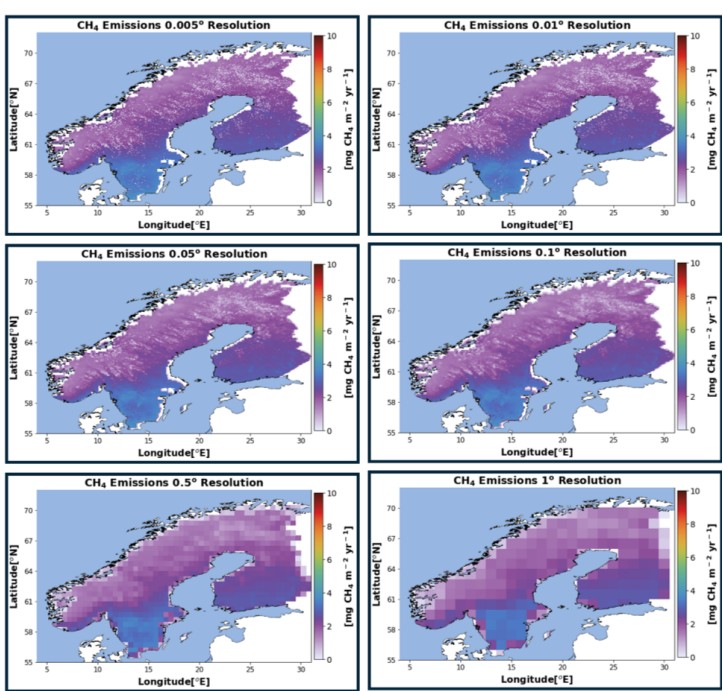


**Figure A- 1: CH₄ emissions for uplands at different resolutions for scenario Sn.2.**

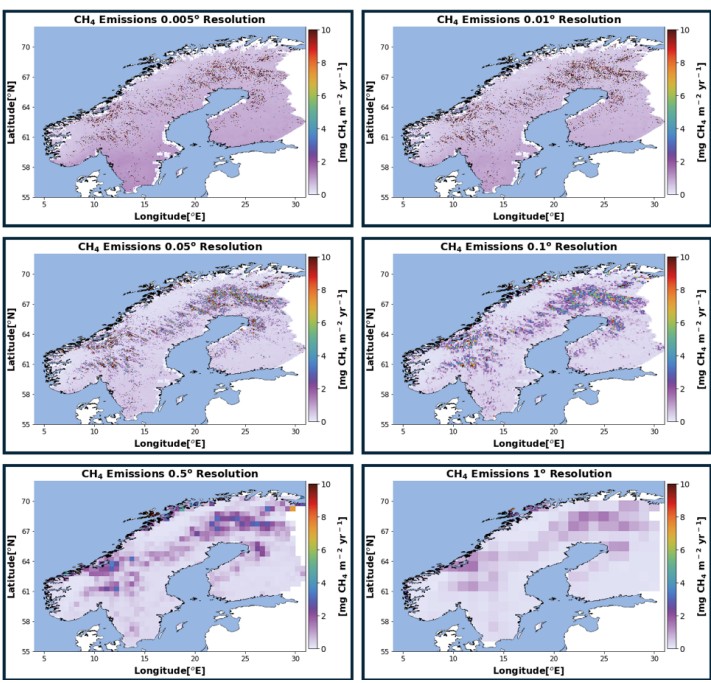

**Figure A- 2: CH₄ emissions for wetlands and uplands at different resolutions for scenario Sn.3.**





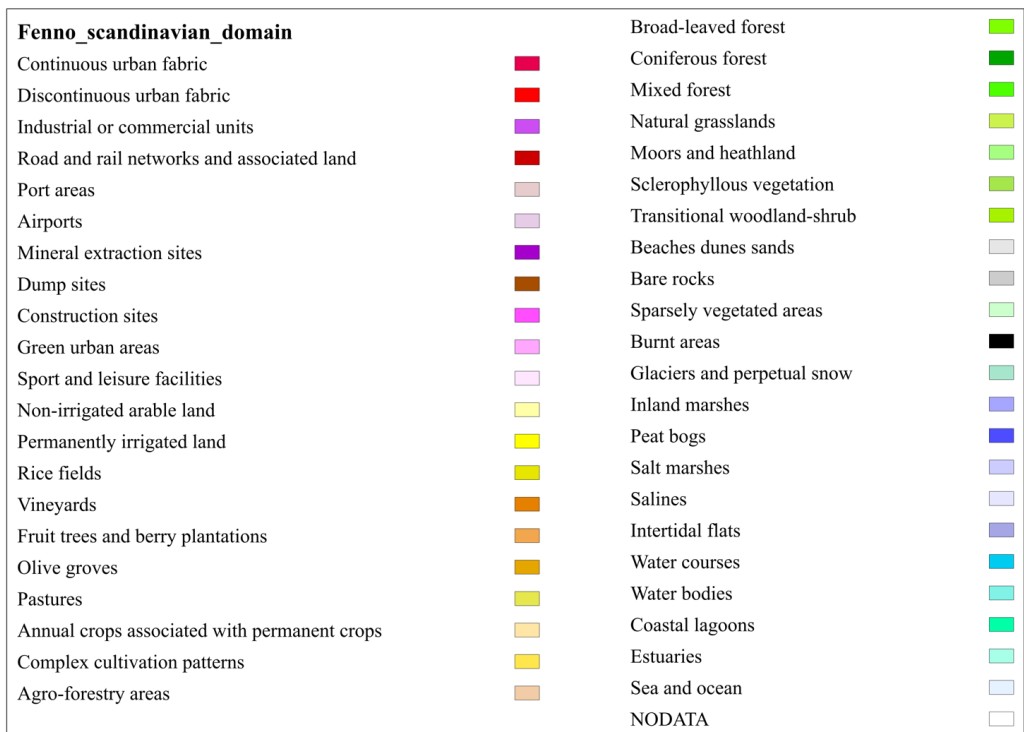

**Figure A- 3: Corine 2018 land cover classification.**



**Appendix B**

**Table B. 1: List of wetland extent maps used for comparisons.**

| Model | Wetland determination scheme | Original Resolution (lon x lat) | Principal references |
|---|---|---|---|
| LPJ-Bern | Prescribed peatlands and monthly inundation. Simulated dynamic wet mineral soils (saturated, non-inundated). | 0.5°x0.5° | Spahni et al. (2011) |
| LPJ-WHyMe | Prescribed peatland extents (Tarnocai et al., 2009) with simulated saturated/unsaturated conditions. | 0.5°x0.5° | Wania et al. (2009a,b, 2010) |
| LPJ-WSL | Prescribed from monthly inundation dataset. | 0.5°x0.5° | Hodson et al. (2011) |
| ORCHIDEE | Mean yearly extent over 1993–2004 period scaled to that of inundation dataset with model calculated intra- and inter-annual dynamics. | 1.0°x1.0° | Ringeval et al. (2010, 2011) |
| SDGVM | Independently simulated extents. | 0.5°x0.5° | Hopcroft et al. (2011), Singarayer et al. (2011) |
| DLEM | Maximal extents from inundation dataset with simulated intra-annual dynamics. | 0.5°x0.5° | Tian et al. (2010, 2011); Xu and Tian (2012) |
| GLWD-3 | Created on the basis of existing maps, data and information, such as the Digital Chart of the World, World Conservation Monitoring Centre (WCMC) and others. | 30 second | Lehner et al. (2004) |
| GIEMS | Satellite based inundated surface data for each month for 15 years (1993-2007) for each pixel on the globe. | 0.25°x0.25° | Prigent et al. (2001, 2007, 2012) |