# Peer review of "The importance of spatial resolution in the modelling of methane emissions from natural wetlands"

_Biogeosciences, 2022_

## Author Comment (AC1)

**Author's response to Reviewer#1 (anonymous)**

General comments:

The goal of the paper is of interest for the scientific community and the approach seems interesting but because the paper is not easy to understand doubt remain on the approach. For instant, some 5 details are missing in sections 2 and 3 that are explained later in the results or discussion sections and reduce the clarity of these sections. For example, in section 3 maps resolutions are provided but not explanation is given on how maps are rescaled or not to be employed for the simulations at various resolution. Then the results description in section 4 is fine but is structured with too many subsections. The content of the result section can stand with no subsections. I believe. While results are 10 consistent with the previous section 1 to 3, the discussion section 5 is disappointing. There is a discussion on the wetland map resolutions and a reanalysis of wetland extend of models employed in

- the WetChimp model intercomparison paper of Melton et al. (2013) and a very small discuss on methane emissions and on the actual simulation results of the paper. It is also disappointing not to have some discussion on the complexity of methane emissions models for example for models
- 15 employed in the WetChimp model intercomparison versus the simple model employed here. The main conclusions of the paper is that wetland distribution is the main uncertainty for methane emissions and it has been demonstrated using a simplified model and conceptual framework. However, this uncertainty has already be addressed and demonstrated in a different manner in the paper of the global methane budget by Saunois et al. (2020) by comparing methane emissions 20 estimated by 13 land surface models using the same wetland map.

We thank reviewer #1 for her/his positive comments, their detailed review and for the constructive recommendations. We respond thereafter to each of their comments.

**Specific comments:**

- Abstract: Line 13 the rang of resolution, from 0.005° to 1° resolution, indicates in the abstract is 25 different than the one in table 3 that range from 0.001° to 1° resolution. Could you explain why? You show results at 0.001°, even though it is employed as a reference, it is still compared to the other runs.

Response: We meant to say that we aggregate from the reference resolution (0.001°) to six resolutions starting from 0.005°. However, the sentence will be revised as "This is done using a high-resolution wetland map (100x100 m2) and soil carbon map (250x250 m2) in combination with a highly simplified CH4 emission model that is coarsened in six steps from 0.001° to 1°.

- Units: Sometimes the resolutions are given in different units than degree such as in line 147, 191 and 256; please make sure that all the units are the same for each variable.

Response: The native PCRG model resolution is in arcmin units. The units of PCRG resolution will be revised as (5 arcmin  $\sim 0.083^{\circ}$ 35

Also, some numbers in the text are formatted using scientific notation, others are not, such as in lines 298-303. It will ease the reading of the paper to have the same format of numbers.

Response: all numbers will be reformatted using scientific notation.

40

- Although, the author sometimes qualifies the methane model by "highly simplified model" I think the model is a "simple model".

Response: "Highly simplified model" is replaced by "simple model".

**45 **Figures and Tables:**

- I believe that Figure 1 and 2 can be merged into one single figure and by adding case 1 next to the content of Figure 1, case 2 next to the content of figure 2 and adjusting the figure caption.

Response: Done.

50 - Figure 3 can be removed, it is not useful to understand the paper. Possibly it can be placed in the appendix or supplementary document.

**Response: Done.**

- Figure 4: My understanding is that you also run Sn 1 at 0.001° resolution which is also you 55 "reference resolution" why does it not appear in Figure 4?

Response: We agree with the reviewer at this point. The reason we choose not to include it because the reader won't see the difference between  $0.001^{\circ}$  and  $0.005^{\circ}$  since both are at very high resolution. We will replace the  $0.005^{\circ}$  figure with  $0.001^{\circ}$  to keep figure 4 in the shape of 4x2 (rows, columns)

Figure 5: the axis labels are not clear what is the right y axis, methane concentration? methane emissions? and the left y axis which ratio is it? Figure captions should describe more the figures. Also, I would advise to modify Sn.1-Sn.3 to Sn.1 to Sn.3 to avoid any misunderstanding such as the difference between Sn.1 and Sn.3

Response: We agree with the reviewer on this point. The figure and caption are modified as below:

Figure 5: Resolution dependence of CH4 emissions for Sn.1 to Sn.3. The right y-axis represents the ratio of the emissions between emissions of the reference resolution (0.001°) to the coarsened resolution step as described in section 2.2. The left y-axis represents the domain integrated annual methane emission.

-Figure 6: please add the resolution for each box diagram in larger characters. Also, axis need labels that describe each axis in addition of the units. In each box diagrams all text should be enlarged except for the boxes with  $K_{CH4}$ .

Response: The resolution of each plot is now mentioned in the title of the plot. The figure is modified as follows:

70

---

## Author Comment (AC2)

**Author's response to Reviewer#2 (anonymous)**

General comments:

This manuscript investigated the importance of spatial resolution in quantifyiing $CH_4$ fluxes from natural wetland. My main issue with this manuscript is that the authors made a lot of simplifications (chose certain values/threhold, and 'randomly' took different datatsets, for instance using reanalysis soil temperature data and then modelled soil mositure at 5 cm) without sound reasons. By taking a lot of unrealistic values, the authors want to demostrate the impacts of resolution on the modelled total methane emissions. There are so many steps/simplifcations which could lead to different results: such as data resampling approach (how did the authors get different resolutions of wetland maps?), soil moisture and soil temperature dataset used (why only top soil layer, and why took two very different datasets?) and also why use the same dataset to calibrate model parameter and evaluate the results.

The authors should really work through their approaches, and thoroughly investigate how different steps/assumption might influence their final results. It is really difficult to understand the simplifcation/approach and the article is lacking in depth and impartiality. Thus, I would not recommend it for publication in its current form.

We thank the reviewer for his/her time and effort to review our manuscript. We acknowledge that rough assumptions have been made, but not necessarily rougher than those used in models that are used to estimate boreal/arctic emissions of methane. The reason for choosing typical values for soil moisture and soil carbon (which may seem just to simplify things even further) is that we came to realize that models, due to their low resolution, effectively use soil moisture and soil carbon values in wetland pixels that deviate significantly from what is measured inside actual wetlands. This is problematic for us since the resolution dependence we study changes with the contrast between wetlands and uplands. So, what may appear is oversimplifications are actually attempts to account for short-comings of existing datasets that might influence the size of the resolution dependence that we estimate. Simplifications also serve the purpose of clearly explaining and isolating the resolution impact that we are after. They are followed by attempts to compute resolution dependences in more refined datasets. Therefore, we effectively do both, simple and detailed, comment on their differences and potential implications.

We are happy to make an additional attempt to further clarify the choices that are made. However, we were surprised that the point about simplifications led to the recommendation to reject our paper. The only argument to reject is the point about simplified datasets, which would have been valid if other choices of existing datasets would obviously have resulted in a significantly different and better estimates of resolution dependence. However, no argumentation along this line is provided. We are of the opinion that paper rejections should be accompanied by thorough argumentation. In this case, the rejection recommendation lacks the supporting argumentation. In our opinion it does not make sense to reject a paper just because a simplified approach has been taken to make a scientific point, unless the approach makes the outcome invalid. For this, in our case, no argumentation is given.

**Specific comments:**

L42-43: Another important process which could contribute to the differences between top-down and bottom-up methods are the details in $CH_4$ transport pathways.

**Response:** Thank you for this suggestion, which we have included in the manuscript.

L128, the unit for the $K_{CH4}$ is not correct. The authors should define the meaning of K values, especially if using $T_0$ equals to 273.15.

**Response:** The unit that is mentioned did not refer to $K_{CH4}$ but to the methane emission. The unit of $K_{CH4}$ has been added to avoid confusion. A clearer definition of $K_{CH4}$ is added too.

L134, Is it true to assume all upland soils take up methane?

**Response:** No, but the sentence that the reviewer refers to does not mention that. It states that upland soils take up methane, which is true in most cases.

L157-158, what do you mean by "network density"? Then in the earlier part of the sentence, it says the reason to choose this study area is due to data availability at a few sites, but then it turns out only 2 sites were monitored. It reads contradictory.

**Response:** It is true that the measurement availability in Scandinavia is relatively high compared with other regions in the Boreal/Arctic zone. However, we agree that "higher network density" suggests something else. We have reformulated this part to avoid contradiction. The sites that were used have by far the longest and most complete measurement record, explaining our choice.

L176: why here comes with scenario 5 now? I did not follow here.

**Response:** Sorry for this mistake, which came from an earlier version. This problem has been fixed.

L176-182, I am not sure I follow the authors' argument to make this type of assumption. 1, why do you need to stick to the highest density values from a dataset that is limited by the peat fractions? 2. It is fine to just assume zero emissions from upland soil, but on L180-192, why this assuption is linked to different soil carbon contents?

**Response:** for the first question: maximum values are used because mixed wetland/upland grid boxes will have lower soil moisture and soil carbon values than the fraction that is covered by wetlands. The typical values that we found in literature for wetlands corresponded reasonably well with the maximum value in the dataset, supporting our approach.

Second point: Zero values were used for uplands so that our simple wetland model only yields methane emissions over wetlands. We agree that this assumption overestimates the soil moisture and soil carbon contrast between wetland and upland, and thereby the resolution dependence, which we address using other scenarios. A sentence was added to explain this more clearly

- L192-193, not sure why you only need to use the top 5 cm soil layer, and which approach did you use it for resampling? It is not clear for me why you introduce another hydrological model for getting soil moisture? Have you evaluated the modelled soil moisture? Why not use the soil moisture data from ERA5?

**Response:** This question is composed of different questions, which we respond to each individually below.

why you only need to use the top 5 cm soil layer?

The study of Shao et al., (2017) showed that the $CH_4$ production potential of the top 5 cm soil layer at different water levels is higher than that of other soil levels.

which approach did you use it for resampling?

**Response:** The resampling is simply a re-gridding. Since the resolution is changed in integer intervals this is straightforward.

It is not clear for me why you introduce another hydrological model for getting soil moisture? Have you evaluated the modelled soil moisture? Why not use the soil moisture data from ERA5?

**Response:**

This study uses the same hydrological model for estimating methane emissions as used in Petrescu et al., (2010) see(https://doi.org/10.1029/2009GB003610)
It has been tested used and evaluated in several studies (e.g. Sutanudjaja et al., 2018) see (https://gmd.copernicus.org/articles/11/2429/2018/).

[Figure]

[Figure]

As shown in the figure above, the PCR-GLOBWB (left) hydrological model has a more sophisticated representation of hydrology than HTESSEL (right) used in ERA-5, which is a reason why we prefer its use over ERA-5.

L 208-210: not clear how the authors obtain the K_CH4 value? What values did you get for KCH4?

**Response:** The $K_{CH4}$ is obtained by 'calibrating' our model to observations. The values are mentioned in the text (line 258:273) and in Figure 6.

Figure 3, where are the color legend?

**Response:** The Legend is in the appendix (Figure A-3). The figure will be revised in the final vision with a wetland distribution map for the study area so that the legend will fit in the figure.

Figure 4, why are there emissions for Sn2 if you only have upland?

**Response:** This question was raised also by RC#1. Here is the answer:

We assume that the answer of this question is fulfilled from line 136 to 141. "In Sn.2 uplands are treated as the wetlands in Sn.1. $CH_4$ oxidation in upland soils may show a resolution-dependence following the logic of section 2.1 also. However, since the upland fraction is generally substantially larger than the wetland fraction at spatial resolutions that are common in global wetland modelling, the sensitivity of the sink to resolution is expected to be less important (see equation 4). The setup of Sn.2 is meant to isolate the impact of the difference between wetland and upland fraction on the resolution dependence, which explains why the method to compute the flux is kept the same"

Figure 6, not clear for me how you can use the same observation data to calibrate K_CH4 and then evaluate the calculated model, please clarify this.

**Response:** This is indeed a valid concern. Nevertheless, it is useful to make this comparison because we only calibrated our model using a single $K_{CH4}$ value, which scales the model output but otherwise does not influence the comparison between model and data. Figure 6 is not meant to evaluate $K_{CH4}$ but to assess the performance of the model given an optimised $K_{CH4}$.